# Comparative Mitogenomics Provides Valuable Insights for the Phylogeny and New DNA Barcodes of *Ganoderma*

**DOI:** 10.3390/jof10110769

**Published:** 2024-11-05

**Authors:** Ti-Qiang Chen, Chi Yang, Xiao-Lan Xu, Lin Yang, Huan-Qing He, Meng-Ting Weng, Zheng-He Ying, Xiao-Kun Shi, Meng-Guang Ding

**Affiliations:** 1Institute of Edible & Medicinal Mushroom, Fujian Academy of Agriculture Sciences, Fuzhou 350014, China; yc113078@163.com (C.Y.); w_mengting@163.com (M.-T.W.); yingzhenghe@126.com (Z.-H.Y.); dmg6703620@yeah.net (M.-G.D.); 2College of Bee Science and Biomedicine, Fujian Agriculture and Forestry University, Fuzhou 350002, China; 3Chengdu Jinxu Biotechnology Co., Ltd., Chengdu 610021, China; 13602583854@163.com; 4Institute of Vegetable Research, Guangdong Academy of Agricultural Sciences, Guangzhou 510640, China; hhq407@sina.com

**Keywords:** *Ganoderma*, mitogenome, collinearity analysis, phylogenetic analysis, candidate genes, DNA barcodes

## Abstract

*Ganoderma* is the most important genus in the family Ganodermataceae; many species have attracted much attention and widely cultivated because of their medicinal values, but so far, not a sequenced mitogenome derived from dikaryon strains has been explicitly recorded. Herein, four novel mitogenomes of commonly cultivated *Ganoderma* (*G*. *leucocontextum* H4, *G. lucidum* G6, *G. sinense* MZ96 and *G. tsugae* SS) were de novo assembled and given detail functional annotations. Collinearity analysis revealed that the four mitogenomes shared 82.93–92.02% similarity with their corresponding reference mitogenomes at the nucleotide level. A total of 15 core protein-coding genes (PCGs), along with *rrn*L and *rrn*S (mtLSU and mtSSU) were chosen as potential candidates for constructing their individual phylogenetic trees. These trees were compared with those derived from the concatenated sequences of 15 core PCGs. And finally, we found that the *atp*9 and *nad*4L were the most reliable markers for the phylogenetic analysis of *Ganoderma* and chosen as standard sequences to generate new DNA barcodes. This finding was further verified by comparing it against almost all available *Ganoderma* mitogenomes in the NCBI, with *Trametes versicolor* (Polyporaceae) and *Rigidoporus microporus* (Meripilaceae) as two outgroups. A total of 52 mitogenomes from three families were highly conserved, with identical gene lengths for *atp*9 (222 bp) and nad4L (267 bp). These genes were capable of distinguish distinctly different various species, which are grouped into separate clades within the phylogenetic trees. The closest related clades (I and II), including at least 30 samples of the three classical taxonomic species (*G. lingzhi*, *G. sichuanense* and *G. lucidum*), differed in only one SNP. The single base mutation rate increased with the evolutionary divergence of the phylogenetic clades, from two to three SNPs in earlier clades (e.g., clade IV containing *G. leucocontextum*) to five to six SNPs in later clades (e.g., clade X containing *G. sinense*). Despite these variations between species, the *atp*9 and *nad*4L genes of *Ganoderma* mitogenomes consistently encoded the same ATP synthase F0 subunit c (73 aa) and NADH dehydrogenase subunit 4L (88 aa). These two genes have been identified as reliable markers of new DNA barcodes, offering valuable insights and contributing significantly to understanding the evolutionary relationships and phylogeny of the *Ganoderma* genus and even the Ganodermataceae family.

## 1. Introduction

Since the initiation of the JGI Fungal Genomics Program [1], at least 22 whole genomes of *Ganoderma* were available in the NCBI databases (https://www.ncbi.nlm.nih.gov/datasets/genome/?taxon=5314, accessed on 12 September 2024), including *G. lucidum* (7), *G. boninense* (4), *G. tsugae* (2), *G. leucocontextum* (2), *G. sinense*, *G. lingzhi*, *G. meredithae* and *G. multipileum*. Notably, only two sequenced strains (*G. lucidum* G.260125-1 and *G. sinense* ZZ0214-1) have been assembled to the chromosomal level [2,3,4], a few sequenced strains have genomes’ BUSCO assessment results, e.g., *G. lucidum* Ling-Jian NO.2, *G. tsugae* CCMJ4178, *G. leucocontextum* DH-8 and *G. leucocontextum* Dai 12418 [5,6,7]. Especially the whole genome of *G. leucocontextum* Dai 12418 had 99.8% complete BUSCOs [7], which was completed under the Fungal Genomes Project Extension (JGI Project ID: 1270834, https://www.ncbi.nlm.nih.gov/datasets/genome/GCA_022813035.1/) and submitted by DOE Joint Genome Institute (6 April 2022).

In addition to the nuclear genome, the mitochondrial genomes of *Ganoderma* species also provide valuable information [8,9]. Mitochondrial genomes have emerged as ideal materials for genomics and evolutionary studies due to their small size, stable gene composition, predominantly maternal inheritance and limited recombination events. These characteristics provide a rich source of molecular markers for phylogenetic and other related researches [10]. Mitochondrial genomes in fungi, similar to those in animals and plants, encode a subset of proteins necessary for mitochondrial function and are inherited maternally. They exhibit high mutation rates and are thus as valuable tools in molecular evolutionary studies [11]. The analysis of mitochondrial genomes can reveal patterns of gene rearrangement, substitution rates and nucleotide composition biases, which are informative in reconstructing evolutionary histories [12]. In summary, the necessity of mitochondrial genome analysis in the study of *Ganoderma* systematics and evolution lies in its potential to unravel the complex evolutionary histories, uncovers conserved and lineage-specific genetic features and provides insights into the biological and pharmacological properties of this valuable fungal resource.

There are 61 records available in the NCBI Nucleotide database for *Ganoderma* (https://www.ncbi.nlm.nih.gov/nuccore/?term=Ganoderma+mitochondrion%2C+complete+genome, accessed on 7 October 2024), with completed sequence lengths ranging from 40,719 bp to 124,588 bp. Among them, 11 records were accessed on the NCBI Reference Sequence Database, and these provisional reference sequences have not been subject to final review, which should be deducted accordingly. The 11 records are all listed below: NC_085596.1, NC_062658.1, NC_037936.1, NC_037937.1, NC_037938.1, NC_026782.1, NC_027188.1, NC_022933.1, NC_021750.1, NC_068513.1 and NC_061293.1 (Identical to OR911496.1, MT843208.1, MH252533.1, MH252534.1, MH252535.1, KP410262.1, KR109212.1, KF673550.1, KC763799.1, OL372261.1 and MN563752.1, respectively). So, there are actually 50 mitogenomes of *Ganoderma*.

It is observed that these *Ganoderma* mitogenomes were almost completed by Illumina sequencing technology, rarely by third generation technologies. Recently, the first dikaryotic genome of *Ganoderma* cultivar Zizhi S2 has been completed using a combination of the PacBio Sequel and Illumina HiSeq sequencing platforms, eventually assembled into a nuclear genome (56.76 Mb, 16,681 genes) with 98.15% of complete BUSCOs, and the No.49 scaffold (85,353 bp) was found to contain the start and end sequences that overlap, which finally led to a complete mitogenome (85,389 bp, GenBank: MN356101) [10]. Recently, the first T2T genome assembly of *G. leucocontextum* strain GL72 has been reported, with a size of 46.69 Mb and complete BUSCOs of 99.7%, and a mitochondrial genome of the size of 89,684 bp was simultaneously assembled [11]. However, this mitogenome cannot be queried currently in the NCBI data but may be available sometime later.

It was reported that the number of protein-coding genes (PCGs) identified in the mitogenomes of five *Ganoderma* species ranged from 22 to 47, and all five mitogenomes contained 14 conserved core PCGs (atp6, atp8, atp9, cob, cox1, cox2, cox3, nad1, nad2, nad3, nad4, nad4L, nad5 and nad6) and one rps3 gene [10]. These 14 conserved genes in the fungal mitochondrial genome have been used as a reliable molecular tool in the study of the phylogenetic relationship [12,13,14].

In this comprehensive study, we have successfully sequenced the mitogenomes of four commonly cultivated *Ganoderma*. Subsequently, these newly sequenced mitogenomes were utilized for an in-depth comparative analysis with the corresponding mitogenomes of related species available in NCBI database. Our primary objective was to meticulously screen suitable core genes, encompassing the 14 PCGs and/or rps3, for phylogeny analysis of *Ganoderma*. Furthermore, we anticipated that this endeavor would unveil innovative and practical insights, thereby facilitating the identification and classification of *Ganoderma* species exhibiting homologous characteristics.

It is well-known that several highly conserved genes, such as nrSSU and nrSSU, *tef* 1-α, *β*-tubulin, *rpb*2, mtSSU and mtLSU, were extensively utilized in the phylogenetic analysis of *Ganoderma* [15,16,17,18,19,20,21]. However, these sequences represented partial CDS (incomplete gene sequences) that were commonly acquired through amplification sequencing, employing the Sanger method. Specifically, the collinearity analysis results of the mitogenome contribute to understanding the phylogeny and evolution of *Ganoderma* and Ganodermataceae, indicating that the gene order of protein and rRNA genes among their mitogenomes is highly conserved [1]. Thus, this study also hopes to mine complete gene sequences from the mitogenomes for molecular identification and molecular evolution.

## 2. Materials and Methods

### 2.1. Background Information of Sequencing Strains

Four sequencing strains were *G. sinense* Minzi 96 [22,23], *G. lucidum* Red Reizhi No.6 [24,25], *G*. *leucocontextum* H4 and *G. tsugae* SS, abbreviated here as G6, BRLA, M96 and SS, respectively. The comprehensive information of these strains was detailed in Appendix A. Notably, the rDNA-ITS2 of *G*. *leucocontextum* H4 (GenBank: PP467490.1) was 100% consistent with the genomic rDNA-ITS2 of *G. leucocontextum* Dai12418 (GenBank: MU404615.1). Both sequences shared precise single-base deletion (G, No.66 bp) when aligned with the ITS2 of *G. leucocontextum* voucher GDGM 40200 (holotype) (Appendix A). Moreover, the cultivated strain SS was originally identified as *G. tsugae* according to the characteristics of the fruiting body and micromorphology of basidiospores (Figure 1).

### 2.2. Sequencing and Assembly of Mitogenomes

The total DNA samples were extracted from dikaryotic mycelium, following a previously reported method [10]. Subsequently, the quality and quantity of these samples were detected by the Nanodrop, Qubit rapid fluorescence accurate quantification and 0.35% agarose electrophoresis, respectively. Utilizing the SQK-LSK109 ligation sequencing Kit, the libraries were constructed. The whole genomes were then comprehensively sequenced on Illumina NovaSeq 6000 sequencing platforms, and detailed statistics of the second-generation sequencing data are provided in Appendix A. For the de novo assembly of the mitogenome, clean data were used for the mitogenome assembly by Norgal v.1.0.0 (https://bitbucket.org/kosaidtu/norgal) [26]. This automated process aligned and exported an overlapping contig with a circle sequence, respectively. Furthermore, GC-Depth analysis was performed on the assembly results, and the GC content and average sequencing depth were calculated without repetition using 500 bp as a window.

### 2.3. Component Prediction and Functional Annotation

The gene component predictions of the assembly results were performed using the MFannot software (https://megasun.bch.umontreal.ca/apps/mfannot/) and then were identified by MITOS [27] and MiTFi (mitochondrial tRNA finder) [28]. To annotate the gene functions, we employed BLAST software in conjunction with an extensive array of protein databases, including the NR-Non-Redundant Protein Database (NR, 2017-10-10 release), Cluster of Orthologous Groups of proteins [29,30] (COG, 2014-11-10 release) and EuKaryotic Orthologous Groups of proteins (KOG), the UniProt Knowledgebase (UniProtKB)/Swiss-Prot [31,32] (2017-07 release), Kyoto Encyclopedia of Genes and Genomes [33] (KEGG, v81), InterProsacn [34] (IPR, v5.31) and Gene Ontology [35] (GO, 2017-09-08 release). The four newly completed mitogenomes with annotation information have been submitted to the NCBI and assigned accession numbers. And the mitogenomic circus maps were constructed with Circos v0.69-6 (http://www.circos.ca), or used the visualize module of MitoZ v2.2 (https://github.com/linzhi2013/MitoZ) calls the Circos software to draw the circus maps.

### 2.4. Collinearity Analyses of Mitogenomes

There are nine genomes as analysis samples: four newly completed mitogenomes, four corresponding reference mitogenomes of *Ganoderma* downloaded from the NCBI data and our previously completed one (MN356101.1). The alignment results (BLASTn and BLASTp) of protein-coding gene sets were proportionally scaled based on the total length and position of each pair of mitogenomes. The reference mitogenomes include *G. sinense*_KF673550.1, *G. lucidum*_HF570115.1 [8], Zizhi.S2_MN356101.1 [10], *G. tsugae*_MH252533.1 and *G. leucocontextum*_MH252534.1 [12]. Collinearity analyses were carried out by MUMmer v3.22 (http://mummer.sourceforge.net/) [36]. Subsequently, 2-dimensional collinearity maps at the nucleotide and amino acid levels were constructed respectively.

### 2.5. Phylogenetic Analysis of Core Genes

Building upon above analysis, 15 core genes (14 PCGs and *rps*3, see Appendix A for details) in the nine mitogenomes were selected as candidate genes for multigene sequence alignment by CLUSTAL [37] (http://www.clustal.org/download/current/clustalw-2.1-win.msi). Subsequently, phylogenetic analyses were conducted both on the entire set of 15 core genes (generating a super-sequences matrix) and on each gene individually, utilizing TreeBesT v1.9.2 (https://mybiosoftware.com/treebest-1-9-2-softwares-phylogenetic-trees.html) and the NJ method and compared to the phylogenetic trees constructed from matrix concatenation, respectively. The selected gene sequences were edited with Notepad++ V 8.6.4 (https://notepad-plus.en.softonic.com/) and aligned with ApE v3.1.3 (https://jorgensen.biology.utah.edu/wayned/ape/) [38] and then aligned to find out the SNPS and InDel variations’ sites within species.

### 2.6. Phylogenetic Analysis of Ribosomal RNAs (rrnL and rrnS)

Two rRNA genes were found in each of the above nine samples, including the newly completed four *Ganoderma* mitochondrial genomes and the previously completed one (MN356101.1), as well as four reference mitochondrial genomes of *Ganoderma* registered in NCBI database. Their introns (Appendix A) were used to construct phylogenetic trees for comparative analysis.

### 2.7. Verify with All Mitogenomes of Ganoderma Available in the NCBI

In order to verify the results of above analysis, the selected gene sequences (*atp*9, *nad*4L) were extracted from all the mitogenomes of *Ganoderma* in the NCBI database (https://www.ncbi.nlm.nih.gov/datasets/genome/?taxon=5314, accessed on 12 September 2024), respectively. As shown in Appendix A, including six mitogenomes of *G. lucidum*, four of *G. sinense*, four of *G. leucocontextum*, 21 of *G. lingzhi* and four of *G. sichuanense*, together with several mitogenomes of other species of *Ganoderma*. As outgroups, *Rigidoporus microporu*s_OR783176.1 [39] and *Trametes versicolor*_ MT479165.1 [40] were selected from Meripilaceae and Polyporaceae, respectively, which belong to the order Polyporales along with Ganodermataceae. Total samples for comparative analysis increased from 9 to 52, and their *atp*9 and *nad*4L selected from the above 15 candidate genes (detailed in Appendix A) were used to construct phylogenetic trees for verification.

### 2.8. New DNA Barcodes Generated from Mitogenome Sequence

The cultivated strain *G. lucidum* G6 has enjoyed widespread commercial cultivation for more than 25 years and remains preserved until now. Based on the previous analysis, *atp*9 and *nad*4l gene sequences in its newly completed mitogenome (PP893276.1) were selected as the reference standard sequence to generate DNA barcodes, and the four-color barcode and QR code were generated under the support of the Chinese herbal medicine DNA barcode system (http://www.tcmbarcode.cn/china/barcodetools/tcmcode/makeimg.php). With these DNA barcodes as reference sequences, we compared samples (species + sequences) that exhibited similar genetic distance in the phylogenetic tree, aiming to uncover SNP sites in them. Moreover, the DNA barcoding was also used to verify their application value by effectively searching online for target genes (*atp*9 and *nad*4L) from the genomes of *Ganoderma* in NCBI databases, for example, BLAST against two genome assemblies of *G. tsugae* (https://www.ncbi.nlm.nih.gov/datasets/genome/?taxon=34467), respectively.

## 3. Results

### 3.1. Assembly Results of Mitogenomes

The clean reads obtained from the Illumina NovaSeq sequencing platform (Appendix A) were used for the mitogenomes assembly of the four sequenced strains, i.e., *G. lucidum* G6, *G*. *leucocontextum* H4, *G. sinense* MZ96 and *G. tsugae* SS. The statistics of the genome assembly for each sample is shown in Table 1. Among that, *G*. *leucocontextum* H4 has the biggest mitogenome of 104,711 bp with a 26.97% GC content, while *G. lucidum* G6, *G. sinense* MZ96 and *G. tsugae* SS are assembled into 65,881 bp, 70,554 bp and 60,320 bp with 26.61%, 26.08% and 26.69% GC contents (Table 1), respectively. The GC content and depth distribution of all the four samples show a gray coniferous scatter plot, without an obvious red area (Appendix A). Since all the four samples were sequenced at great depths (3000, 4100, 5000, even up to 11,000), most of the sequencing data are nuclear genomes while the mitochondrial data are relatively small, the reads coverage rate reaches up to 100% (Table 1).

### 3.2. Features of Four Novel Ganoderma Mitogenomes

The composition and information of mitogenomes showed that the largest one (*G*. *leucocontextum* H4) encoded 67 genes, and the other three encoded 40, 43 and 46 genes, respectively (Table 2). These four novel mitogenomes all derive from the dikaryon strains of *Ganoderma*, with annotation rate of 93.02–97.50%. They entirely contained two ribosomal RNAs (*rrn*L, *rrn*S), 27 transfer RNAs, 14 genes involved in respiratory chain complexes (*atp*6, *atp*8, *atp*9, *cob*, *cox*1, *cox*2, *cox*3, *nad*1, *nad*2, *nad*3, *nad*4, *nad*4L, *nad*5 and *nad*6) and one ribosomal protein gene (*rps*3). In the BRLZ mitogenome, there were three *trn*M-CAT copies, three different *trn*T (*trn*T-TAG, *trn*T-TGT and *trn*T-GGT) with the same anticodon (tRNA-Thr) and two different *trn*S (*trn*S-TGA and *trn*S-GCT) with the same anticodon (tRNA-Ser). In the G6 and MZ96 mitogenomes, there were three *trn*M-CAT copies and two different *trn*S (*trn*S-TGA and *trn*S-GCT) with the same anticodon (tRNA-Ser). The SS mitogenome contained three *trn*M-CAT copies and two different *trn*S (*trn*S-TGA and two *trn*S-GCT copies).

The genomic circle map (Figure 2) revealed that most genes on the outside of the circle were located on the positive strand, and only a few genes (MZ96: *orf*217, *orf*102 and *orf*507; S2: *orf*497, *orf*236 and *trn*R; G6: *orf*568, *orf*283, *orf*109 and *orf*396 and SS: orf568, orf109, orf283 and orf396) on the inside of the circle were located on the negative strand. In contrast, most genes of BRLZ (*G*. *leucocontextum* H4) on the inside of the circle were located on the negative strand, and only five genes (*orf*349, *orf*506, *orf*419, *atp*6, *orf*271 and *orf*235) were located on the positive strand. The above four complete mitogenome sequences with annotation information have been submitted to NCBI and assigned the GenBank accession number, i.e., PP790945.1 (BRLZ), PP893276.1 (G6), PQ301463.1 (M96 or MZ96) and PP860909 (SS).

Furthermore, there were several longer IGS regions. For example, the IGS_2365 bp of *orf*218-*trn*P (38641..41005) and IGS_2313 bp of *orf*254-*cox*3 (97457..102057)of BRLZ, the IGS_2705 bp of *rrn*S-*nad*5 (55728..58432), IGS_2445 bp of *orf*109-*trn*S (14348..16792) and IGS_2742 bp of *orf*319-*orf*568 (8985..11726) in G6, the IGS_2705 bp of *rrn*S-*nad*5 (5978..8682), IGS_2343 bp of *orf*109-*trn*S(25234..27576) and IGS_2741 bp of *orf*319-*orf*568 (19872..22612) in SS, the IGS_4665 bp of *orf*270-*orf*312 (5241..9905), IGS_2160 bp of *trn*R_TCT-*orf*100 (21417..23576) and IGS_3296 bp of *orf*442-*orf*277 (24019..27315) in MZ96, etc. These were also visually presented in the genomic circus maps (Figure 2).

### 3.3. Gene Function Annotation of the Mitogenomes

According to the annotation results, 64 genes were identified in the biggest mitogenome of *G*. *leucocontextum* H4, and 40, 43 and 46 genes in the other three mitogenomes of *Ganoderma*, accounting for 95.52%, 93.02%, 93.47% and 97.50% of the total genes, respectively (Appendix A). Among the seven databases used in annotation, NR, IPR and GO provided high annotation rates, and all annotated a large number of genes (see Appendix A for details). The statistical results of the three-classification annotation of the GO database were shown in Figure 3.

All the four mitogenomes (BRLZ, G6, SS and M96) contained considerable number of ORFs (Table 2). Among those, 18, 11, nine and 17 ORFs showed identity to the homing endonuclease with the LAGLIDADG motif, respectively. It is worth noting that up to 26 of total 44 ORFs in our previously submitted mitogenome of the *G. sinense* strain Zizhi S2 (MN356101.1, 85,389 bp) revealed endonuclease with the LAGLIDADG motif. There are eight, two, two, three and four ORFs endonuclease with the GIY motif in the above mitogenomes (BRLZ, G6, SS, M96 and Zizhi S2), respectively.

### 3.4. Collinearity Analyses at the Nucleotide and Amino Acid Sequence Level

The collinearity analysis revealed that the mitogenomes of the sequenced strain M96 and the previously sequenced strain Zizhi S2 (MN356101.1) were 84.64% and 82.93%, respectively, similar to the reference mitogenome of *G. sinense* (KF673550.1) at nucleotide level (Table 3 and Figure 4), while 93.70% and 96.64 (median 99.70% and 99.38%) at amino acid level (Table 3 and Figure 5). Both cultivated strains M96 and Zizhi S2 of *G. sinense*, matched 30 and 31 genes to the interlinked genes in the reference mitogenome of *G. sinense*_KF673550.1, including all 13 PCGs (*atp*6, *atp*8, *atp*9, *cox*1, *cox*2, *cox*3, *nad*1, *nad*2, *nad*3, *nad*4, *nad*4L, *nad*5 and *nad*6). The details are shown in Appendix A (Sheet 1 and 2) and Appendix A (Sheet 1 and 2).

Moreover, the collinearity analysis displayed 92.02% similarity between the mitogenomes *G. lucidum* G6 (PP893276.1) and *G. lucidum*_HF570115.1 and 100% similarity between *G. tsugae* SS (PP860909.1) and *G. lucidum*_HF570115.1 at nucleotide level (Table 3, Figure 6). At amino acid level, the identity was nearly 100% (Table 3, Figure 7). The mitogenomes of both the cultivated strains G6 and SS (PP893276.1, PP860909.1) had 16 genes with 100% percent matched to the interlinked genes in the reference mitogenome of *G. lucidum*_HF570115.1, also including the 14 PCGs and one extra gene(*rps*3). The details are shown in Appendix A (Sheet 3 and 4) and Appendix A (Sheet 3 and 4).

### 3.5. atp9 and nad4L Are More Reliable for Phylogenetic Analysis

The 15 core genes were identified including ATP synthase subunits (atp6, *atp*8 and *atp*9), cytochrome oxidase subunits (*cox*1, *cox*2 and *cox*3), NADH dehydrogenase subunits (*nad*1, *nad*2, *nad*3, *nad*4, *nad*4L, *nad*5 and *nad*6) and the ribosomal protein S3 (*rps*3). The statistical results of identification and coding sequence length are shown in Table 4. Among those, the lengths of *cob*, *cox*1, *nad*1, nad2, nad4, nad5 and *rps*3 genes (1000–2000 bp) are relatively longer, while the *atp*6, *cox*2 and *cox*3 have medium lengths with some range of variations (750–774, 727–765 and 684–813 bp, respectively), and *nad*3 and *nad*6 varied more (from 360 bp to 1302 bp and from 612 bp to 2175 bp, respectively). Furthermore, there were three genes with shorter sequences, i.e., *atp*8_159 bp, *atp*9_222 bp and *nad*4L_*267* bp, among which better selection may exist. The phylogenetic trees constructed from each candidate gene were compared and matched to the phylogenetic tree constructed from 15 core genes.

Firstly, the nad5 gene was excluded due to its long sequence and numerous SNP/InDel sites. Secondly, the shortest gene (*atp*8_159 bp) was also excluded because *G. sinense*, *G. leucocontextum* and *G. tsugae* could not be well distinguished and clustered into a large branch in the *atp*8-phylogenetic tree (Figure 8a), which was clearly inconsistent with the phylogenetic tree constructed with the 15 core genes (Figure 9a). Sequence alignment also indicated that only two SNP sites were presented in the six samples of the above three different species (Figure 8b). And then, other genes with moderate or longer sequence lengths (*atp*6, *cob* and *cox*1, Appendix A; *cox*2, *cox*3 and *nad*1, Appendix A; *nad*2, *nad*3 and *nad*4, Appendix A; *rps*3, Appendix A) are consistent with the clustering results of phylogenetic trees constructed with 15 PCGs union (Figure 9a) but were not recommended for candidate genes as new DNA barcodes.

Finally, it was clear from Figure 9 that the phylogenetic tree constructed with the other two shorter genes (*atp*9_222 bp and *nad*4L_*267* bp) were all consistent with the 15 PCG-phylogenetic tree (Figure 9a). This indicated that *atp*9 and *nad*4L were more reliable for phylogenetic analysis of *Ganoderma*, which were ideal candidate genes as new DNA barcodes.

### 3.6. rrnL and rrnS Not Suitable for Phylogenetic Analysis

It is shown that the phylogenetic trees constructed with *rrn*L and *rrn*S genes (Appendix A) are significantly different from the phylogenetic tree constructed with the 15 core genes (Figure 9a). For example, the homologous *G. leucocontextum*_MH252534.1 and *G. leucocontextum* H4 (BRLZ) do not cluster together but appear separately on two small clades (Appendix A). In the above Appendix A, it is noted that the number of their introns ranges from one to four, with total lengths of 2500–3200 bp and 1800–2100 bp (*rrn*L and *rrn*S), respectively. Moreover, we also noticed that the *rrn*L and/or *rrn*S genes could not be predicted or annotated completely in some mitogenomes of *Ganoderma* due to the local sequence quality without correction. This leads to a reduction in the number of effective samples (rRNA genes in mitogenomes, mtLSU and mtSSU). Therefore, the *rrn*L and *rrn*S genes in mitogenomes were not recommended for phylogenetic analysis, particularly when considering incomplete rRNA gene sequences (by the Sanger method).

### 3.7. Verified by 52 Mitogenomes of Ganoderma Available in the NCBI

All the 52 samples from the three families (Appendix A) were highly conserved with the same gene length (*atp*9_222 bp and *nad*4L 267 bp), all of which encoded the same ATP synthase F0 subunit c (73aa) and NADH dehydrogenase subunit 4L (88aa). They can be well distinguished from the different species, clustered within large or small (even a single) different 15 clades by genetic relationships, as shown in Figure 10. Among those, the largest one (clade I) contained 29 samples, and *G. amboinense_*ON720161.1 was clustered in with exactly the same *atp*9 and *nad*4L gene sequences.

Meanwhile, the newly completed “*G. tsugae*” strain SS_PP860909.1 was also clustered in this large clade together with three samples of *G. sichuanense* (MH252531.1, MW752413.1 and MW752415.1) because of the same *atp*9 and *nad*4L gene sequence. And the clade II contained newly completed *G. lucidum*_PP893276.1 and two samples of *G. lingzhi* (MT843214.1 and MT843218.1) due to an SNP site (A→G, at the 69th base) between their *atp*9 gene in all 29 samples and the nearest clade I, while the *nad*4L gene is 100% identical and highly conserved within species.

As showed in Figure 10, the clade IV contained another newly completed mitogenomes: *G. leucocontextum* strain H4_ PP790945.1, *G. tsuga*e strain s90_MH252533.1, *G. leucocontextum* strain s116_MH252534.1 and *G. leucocontextum* strain Dai2418_PP212909.1 (reverse). These four mitogenomes’ sizes ranged from 88,194 bp to 104,711 bp. Their *atp*9 gene sequences were 100% identical, but five SNP sites only existed between the *atp*9 genes of the first two samples and the last two samples (see Appendix A). As showed in Figure 10, the clade X also contained OP453745.1_*G. sinense* and MZ96_PQ301463.1 (newly completed), KF673550.1_*G. sinense* and MN356101.1_Zizhi S2; these four mitogenomes sizes ranged from 70,554 bp to 86,451 bp. There are generally two to three SNPs between their *atp*9 and *nad*4L gene sequences within the species (see Appendix A). This indicated that the single base mutation rate of *Ganoderma* species was increased with the evolutionary clades of the phylogenetic tree.

### 3.8. Multiple Uses of DNA Barcodes

DNA barcodes were generated under the support of the Chinese herbal medicine DNA barcode system, as shown in Figure 11. The information can be obtained by scanning the four-color barcodes and QR code.

#### 3.8.1. Measure the Relationship Between Ganoderma

From the phylogenetic tree constructed with the 52 samples (Figure 10), it can be seen that at least 28 samples of three classical taxonomic species (*G. lingzhi*, *G. sichuanense* and *G. lucidum*) in the clade I, due to their identical *atp*9 and *nad*4L gene sequences and can be classified into one complex species. Of course, there was still a sample *G. amboinense*_ON720161.1 in the clade I that required further analysis with additional samples to make a final conclusion, such as treating it as a synonym.

In clade II, there were three samples with one SNP site. Among those, the newly completed mitogenome (GenBank: PP893276.1) of *G. lucidum* G6 were selected as the standard sequence of new DNA barcodes. In clade III, only one sample. i.e., *G. pseudoferreum*_PP778499.1, with four SNPs in *atp*9 and one SNP site in *nad4*L were observed, compared to the standard sequence of DNA barcodes.

In clade IV, the four samples *G. leucocontextum*_PP790945.1, *G. tsuga*e_MH252533.1, *G. leucocontextum*_MH252534.1 and PP212909.1, compared to the standard sequence of DNA barcodes, three and six SNP sites existed in their a*tp*9 and nad4L genes, respectively.

#### 3.8.2. Measure the Relationship of Outgroup Species of Polyporaceae and Meripilaceae

As an ideal outgroup of *Ganoderma* (Ganodermataceae), *Trametes versicolor* (Polyporaceae) was relatively close to *Ganoderma.* The atp9 gene of *T. versicolor* (MT479165.1_23485..23706) and G. *lucidum* (PP893276.1_40924..41145) both encoded ATP synthase F0 subunit c (mitochondrion) consisting of 73 aa. Despite having 21 SNPs in their nucleic acid sequences, their encoding produces (QPF23604.1 vs. XBQ63844.1) were incredibly 100% consistent, as showed in Appendix A. Meanwhile, the *nad*4L gene of *Trametes versicolor* (MT479165.1_36350..36616) was reversed compared to that of G. *lucidum* (PP893276.1_62816..63082), with both encoding NADH dehydrogenase subunit 4L (mitochondrion) consisting of 88 aa. In fact, there are 14 SNPs and two single base InDel sites in nucleic acid sequences, but their encoding products (QPF23609.1 vs. XBQ63859.1) were incredibly 100% consistent, as showed in Appendix A. These two InDel sites appeared at the 46th to 51st base (ATAT-TA vs. AT-TCTA), encoding the same two amino acids (IL, i.e., isoleucine and leucine).

Clade XV at the top of the phylogenetic tree is another more distant outgroup (*Rigidoporus microporus*), which showed dozens of SNPs and InDel sites compared to the reference standard sequence. There were at least 28 mismatches (SNPs), 33 gaps (total bases of InDel), 29 mismatches and 20 gaps between their *atp*9 and *nad*4L gene sequences and the standard sequence of DNA barcodes (Figure 11), respectively. These validated results confirm that *atp*9 and *nad*4L gene sequences were novel DNA barcodes and the ideal gene sequences for phylogenetic studies.

## 4. Discussion

### 4.1. Comparative Mitogenome Analysis Promoted the Phylogeny of Ganoderma

Mitochondrial genomes with relatively small and faster evolutionary rates than nuclear genomes have been successfully used in evolutionary biology and phylogenetic studies and are commonly used to assess the interrelationships among different genera of fungi, different species and different strains of the same species [41]. It was reported that concatenated sequences of *atp*8, *cob*, *cox*2, *cox*3, *nad*2, *nad*3, *nad*4, *nad*5 and *nad*6 genes were selected from 57 mitogenomes of Agaricomycete and used to determine the phylogenetic relationships among seven classes of fungi. In the phylogenetic divergence tree, eight samples of *Ganoderma* (*G. sinense*, *G. lucidum*, *G. tsugae*, *G. leucocontextum*, *G. calidophilum* and *G. applanatum*) clustered into a small branch, while three adjacent samples of *Trametes* clustered into a slightly bigger branch. Furthermore, they clustered with other species of Polyporales together in a large branch [42]. Comparative mitogenome analysis indicated that gene orders in some families of Basidiomycota were highly variable, and large-scale gene rearrangement events commonly occurred in generic level, such as *Cantharellus*, *Ganoderma*, *Lyophyllum*, *Pleurotus* and *Russula* [43,44,45]. The combined mitochondrial gene set was reliable molecular markers for analyzing the phylogenetic relationships of Basidiomycota. Significant genome rearrangements and frequent intron gain/loss events occurred in the mitogenomes of many species of basidiomycete, including *Ganoderma* [12,13].

The evolution relationship between three newly assembled and annotated mitogenomes (*G. resinaceum*_G21, *G. sichuanense*_G22 and *G. flexipes*_G31) and seven *Ganoderma* species of in GenBank (*G. lingzhi*, *G. sinense*, *G. tsugae, G. leucocontextum*, *G. calidophilum*, *G. meredithae a*nd *G. applanatum*) was reported. These were clustered into two different clades with two outgroups in the PCG-phylogenetic tree. Among those, *G. sinense* and *G. calidophilum* were the closest relatives while *G. leucocontextum* had the furthest genetic relationship [46]. In another subsequent report, a total of 19 *G. lingzhi* mitogenomes were assembled and analyzed, combined with three mitogenomes of *G. lingzhi* from GenBank. The results confirmed that the 14 common protein coding genes were highly conserved, although their mitogenomic completed sequence ranged from 49,233 bp to 68,367 bp. And the phylogenetic analysis showed that these *G. lingzhi* strains gathered with high support, and those with the same intron distribution law had closer clustering relationships [47]. In this study, the analyzed samples were expanded to 52 mitogenomes in the NCBI database, including six records of *G. lucidum*, four records of *G. sichuanense* and 2*2* records of *G. lingzhi*, which were clustered into one large and one small clade in the phylogenetic tree (Figure 10, clades I and II) and revealed high conservatism within species.

### 4.2. Collinearity Analysis Actively Plays a Significant Role in the Classification and Identification of Ganoderma

Mitogenomes and their core genes have been extensively providing valuable insights and contributing to the understanding of the evolutionary and phylogeny of *Ganoderma* and Ganodermataceae [8,9,12,13,14]. In general, the collinearity analysis at nucleic acid level can show sequence insertion, deletion and other information than that at amino acid level, which helps to understand the structural variation of the mitogenome between strains during evolution, and the change of the location of gene clusters with similar functions in different strains. According to the collinearity analysis results in this study, it was confirmed that the mitogenomes of the same species should exhibit more than 82% identity at nucleic acid level. For example, 84.64% identity between *G. sinense* M96 and *G. sinense*_KF673550.1, 82.93% identity between Zizhi.S2_MN356101.1 and *G. sinense*_KF673550.1 (Table 3) and 92.02% identity between *G. lucidum* G6_PP893276.1 and *G. lucidum*_HF570115.1 (Table 3).

It is noteworthy that the collinearity analysis of the mitogenomes of “*G. tsugae*” SS_ PP860909.1 and *G. lucidum*_HF570115.1 exhibited a 100% identity at the nucleic acid level. This finding implies that the cultivated strain SS may represent a distinct species from *G. tsugae*, despite its initial classification as *G. tsugae* based on morphological taxonomy. The detailed characteristics of the cultivated fruiting body and micromorphology of basidiospores are presented in Figure 1. Moreover, it is also distinctive on the phylogenetic trees (Figure 8 and Figure 9, Appendix A), this strain SS is also clustered together with *G. lucidum* G6_PP893276.1 and *G. lucidum*_HF570115.1. For *G. lucidum* or *G. lingzhi*, there were enough samples already available for future studies.

There is only one mitogenome (i.e., MH252533.1) of *G. tsugae* in the NCBI database at present (*Ganoderma tsugae* mitochondrion—Nucleotide—NCBI (nih.gov), accessed on 12 September 2024). In this study, a nearly complete mitogenome sequence of *G. tsugae* CCMJ4178 (GenBank: JAHRBS010000017.1) have been successfully identified within the whole genome shotgun sequence (GCA_024305745.1) through the strategic application of DNA barcoding. However, this prospective mitogenome still necessitates further assembly polishing. It is known that the monokaryon strain CCMJ 4178 is a protoplast derived from the wild dikaryotic *G. tsugae* strain CCMJ2475 [5].

### 4.3. Novel DNA Barcodes from Fungal Mitogenomes

Back in 2003, Hebert et al. established that the mitochondrial gene encoding cytochrome oxidase I (COI) can serve as the core of a global bio-identification system for animals. They argued that the sole prospect for a sustainable identification capability lies in the construction of systems that employ DNA sequences as taxon “barcodes”. The concept of DNA barcodes uses one or several paragraphs of standard DNA sequences to achieve the purpose of quickly and accurately identifying animal, plant and fungal species, just like supermarkets use barcodes to distinguish thousands of different products [48].

And later, Kress et al. (2005) further proposed that three criteria must be met for selecting a sequence as DNA barcodes: (1) a large-enough degree of interspecies variation; (2) a short-enough sequence length to facilitate DNA extraction and amplification and (3) the existence of conserved sites to facilitate the design of universal primers [49]. In the following years, extensive analysis and comparison of seven candidate gene sequences (*psb*A-*trn*H, *mat*K, *rbc*L, *rpo*C1, *ycf*5, ITS2 and ITS) were performed in more than 6600 samples belonging to 4800 species from 753 distinct genera. The results indicated that the rate of successful identification with the ITS2 was 92.7% at the species level, and the ITS2 sequence of nuclear ribosomal DNA was highly recommended as the standard DNA barcode of medicinal plants [50]. The ITS2 could be used as a new universal barcode to identify a wider range of animal, plant and fungi taxa. A comprehensive DNA barcode system for identifying plant-based Chinese medicinal materials was proposed, utilizing the ITS2 sequence as the primary marker and the *psb*A-*trn*H as a supplementary marker. In contrast, the DNA barcode identification system for animal-based Chinese medicinal materials employs the COI sequence as the primary marker, with the ITS2 sequence as the auxiliary [51].

In recent years, DNA barcodes have become an important method for fungal systematics, which is widely used in species identification, systematic classification and genetic structure analysis of the population. Among them, the rDNA-ITS sequences, especially the ITS2 region, are universally recognized and used. For most species of *Ganoderma*, ITS is a reliable distinguishing sequence, but for a few species, such as *G*. *leucocontextum, G. tsugae*, *G. lucidum* and *G. lingzhi*, the ITS sequence difference is very small. In addition, the ITS sequence available in the GenBank database exhibits considerable variability in terms of length and quality [52]. Therefore, this study turned to finding suitable gene sequences for new DNA barcodes from the mitogenome sequence of *Ganoderma* and confirmed that two conserved genes (*atp*9 and *nad*4L) with sufficiently short sequence lengths and sufficiently large interspecies variations are ideal candidates for DNA barcodes.

For nearly two decades, molecular phylogenetic studies primarily relied on data derived from one or a few genes, typically obtained through PCR amplification and Sanger sequencing methodologies. In the era of genomics, the advances in high throughput sequencing technologies were observed, and the cost of second-generation sequencing is close to the predecessor. Gene sequence data for phylogenies analysis are abundant in genome databases, which can be derived either from gene predictions based on genome sequences (even from draft-quality genomes) or from transcriptomes generated by sequencing libraries derived from mRNA [53]. Recently, we completed a high-confidence binuclear genome based on comparative genomics analysis, which provided rich genome polygenic sequence evidence for *Ganoderma* species identification and used the newly discovered *fip* gene for phylogenetic studies, providing valuable insights into the molecular evolution of *Ganoderma* [54]. In the future, we can also use the novel DNA barcodes established in this study to mine more and more genetic data for phylogenetic studies on *Ganoderma* and even Ganodermataceae.

## 5. Conclusions

Comparative genomics analysis with four newly completed mitogenomes of the commonly cultivated dikaryotic strains of *Ganoderma* confirmed that the corresponding mitogenomes between the same species shared more than 82% similarity at nucleotide level. The *atp*9 and *nad*4L genes were successfully selected from 15 candidate genes by phylogenetic analyses, the SNP and InDel variations in the nucleotide sequence were sufficient to distinguish different species of *Ganoderma*. The high conservation and polymorphism were verified by almost all available *Ganoderma* mitogenomes in the NCBI database, and both genes were identified as robust markers for novel DNA barcodes.

## Figures and Tables

**Figure 1 jof-10-00769-f001:**
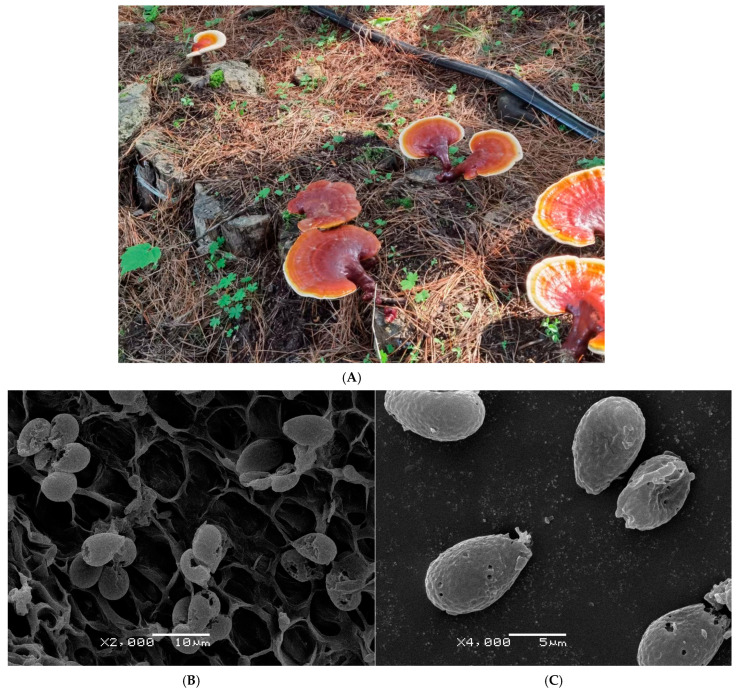
Fruiting bodies and basidiospores’ morphology of “*G. tsugae*” strain SS (**A**): Log-cultivated fruiting bodies under the forest locally in Changbai Mountain (Jilin Province, China), which are very similar to *G. tsugae* in morphological characteristics; (**B**): Basidiospores in the hymenium of a fruiting body (by JEOL JSM-6380LV scanning electron microscope observation); (**C**): Basidiospores scattered on the surface of a mature fruiting body.

**Figure 2 jof-10-00769-f002:**
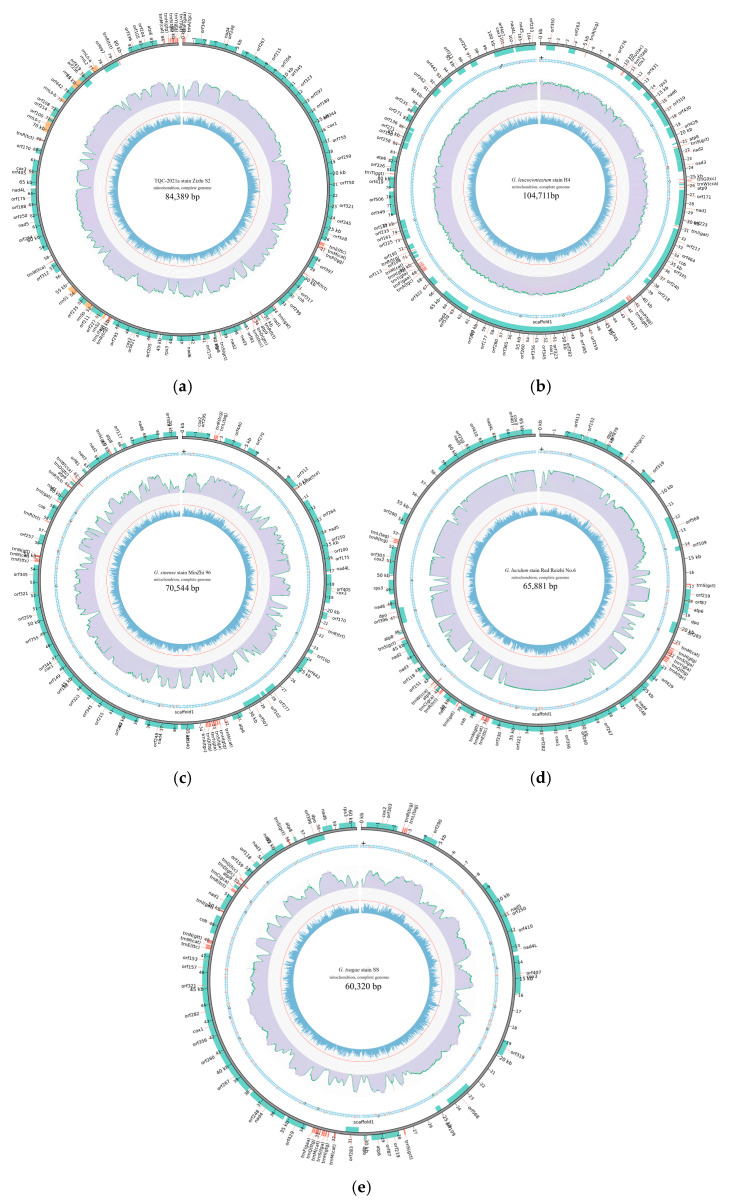
Mitogenome circle maps (a): Zizhi S2 (MN356101); (**b**): *G. sinense* MZ96 (PQ301463.1); (**c**): *G*. *leucocontextum* H4 (PP790945.1), (**d**): *G. lucidum* Red Reizhi No.6 (PP893276.1); (**e**): *G. lingzhi* SS (PP860909.1). Note: From the outside to the inside: the first circle is the information of the genome groups and components (genes, tRNA, rRNA), the outside of the circle indicates the positive chain of the genome, the inside indicates the negative chain of the genome, the red specimen tRNA, yellow represents rRNA, green represents genes; The second circle is: Base track (Only Zizhi S2 did not draw the base track, due to the adoption of PacBio sequencing platform [10] different from this study); The third circle is: the abundance of second-generation sequencing reads was compared to the genome; The fourth circle is: genome GC content; In the middle is the sequencing strain name and genome length information.

**Figure 3 jof-10-00769-f003:**
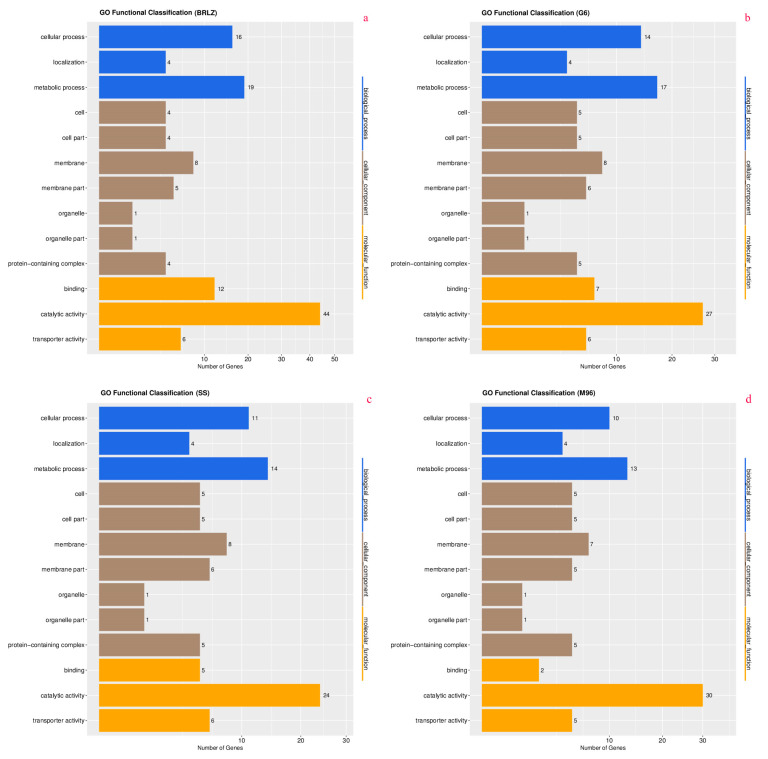
Histograms of GO function annotation distribution (**a**): *G*. *leucocontextum* H4; (**b**): *G. lucidum* G6; *(***c**): *G. tsugae* SS; (**d**): *G. sinense* MZ96.

**Figure 4 jof-10-00769-f004:**
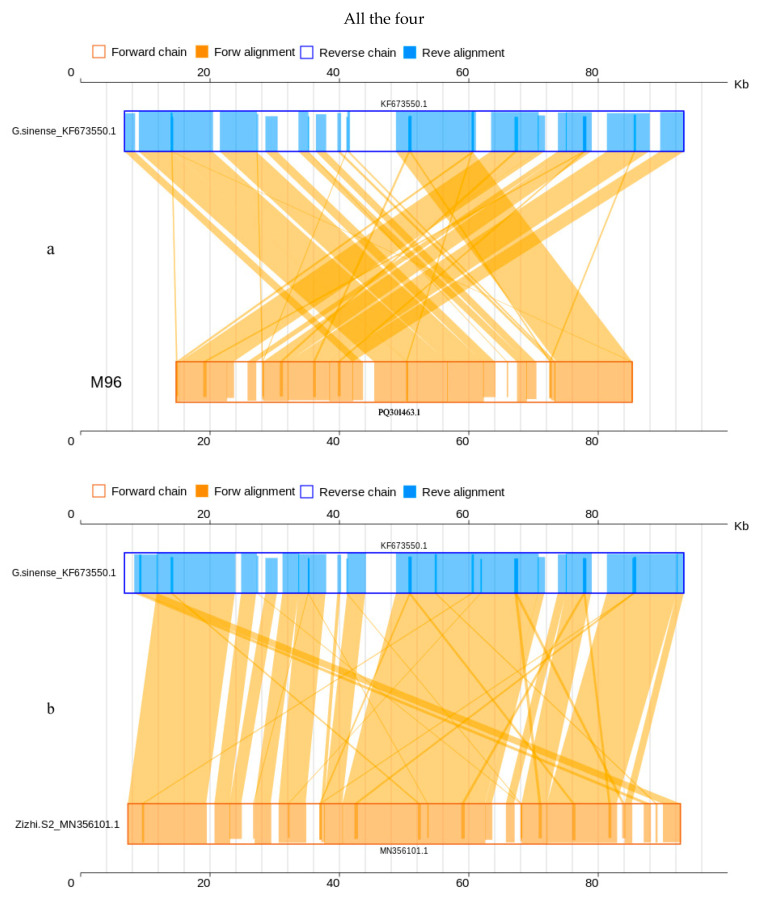
Two-dimensional collinearity maps of mitogenomes at nucleic acid level. The above represents the reference mitogenome sequence of *G. sinense*_KF673550.1, and the below represents the mitogenome of *G. sinense* MZ96 and Zizhi S2_MN356101.1.

**Figure 5 jof-10-00769-f005:**
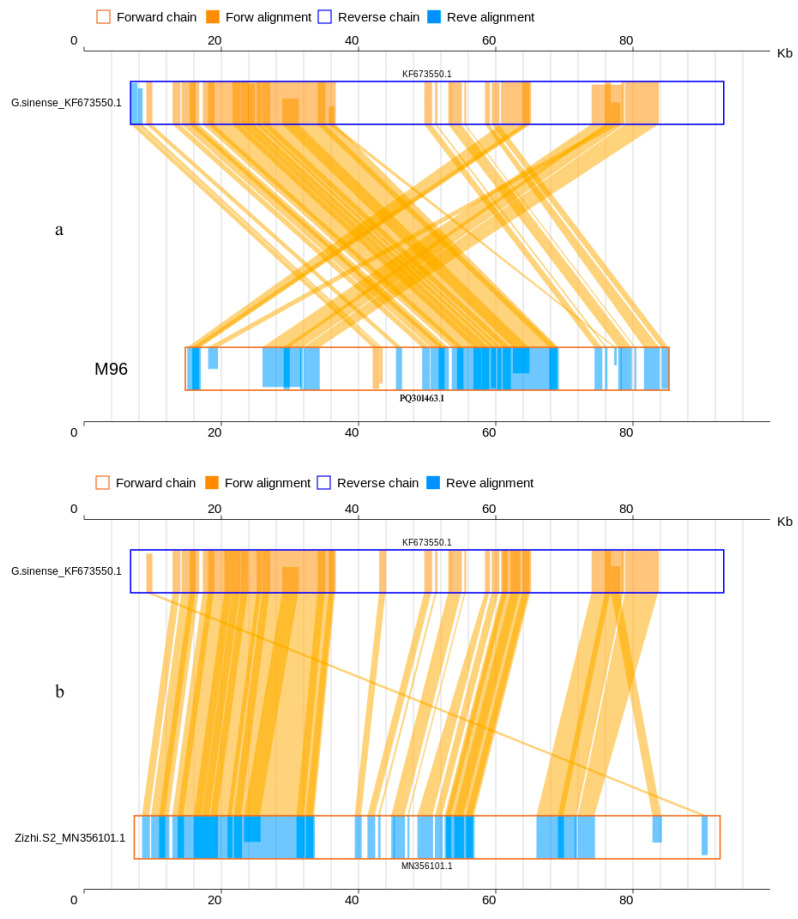
Two-dimensional collinearity maps of mitogenomes at amino acid level. The above represents the reference mitogenome sequence of *G. sinense*_KF673550.1, and the below represents the mitogenome of *G. sinense* MZ96 and Zizhi S2_MN356101.1. Other instructions are the same as above.

**Figure 6 jof-10-00769-f006:**
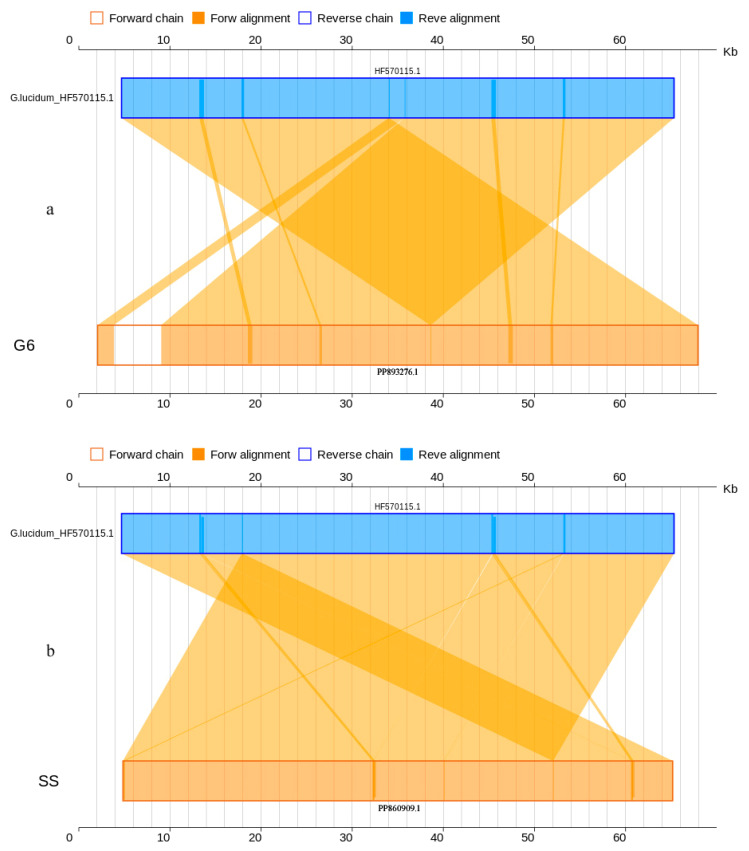
Two-dimensional collinearity maps of mitogenomes at nucleic acid level. The above represents the reference mitogenome sequence of *G*. *lucidum*_HF570115.1, and the below represents the mitogenome of *G. lucidum* G6 and “*G. tsugae*” stain SS. Other instructions are the same as above.

**Figure 7 jof-10-00769-f007:**
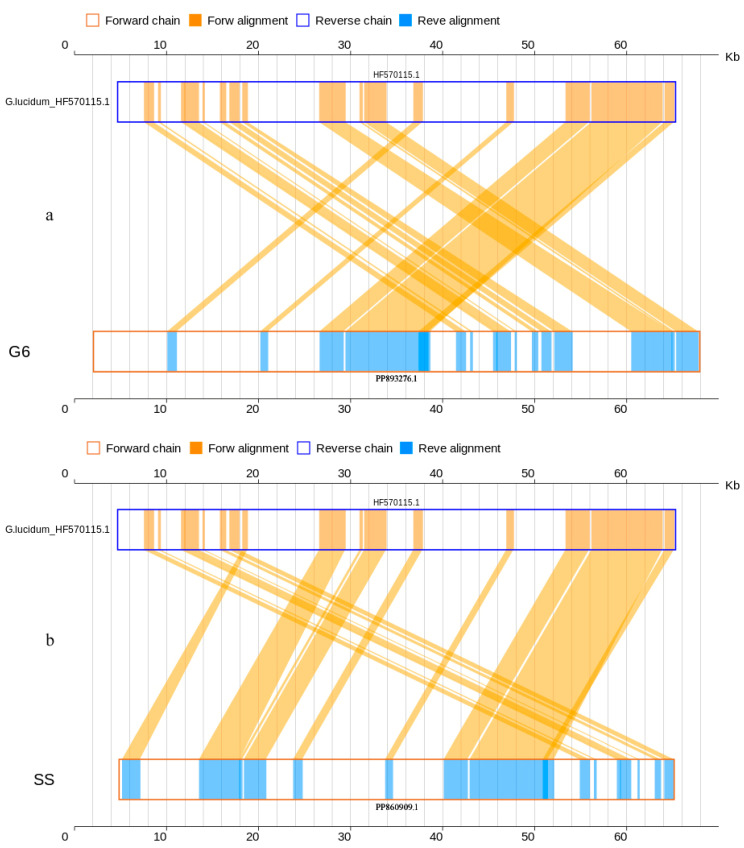
Two-dimensional collinearity maps of mitogenomes at amino acid level. The above represents the reference mitogenome sequence of *G*. *lucidum*_HF570115.1, and the below represents the mitogenome of *G. lucidum* G6 and “*G. tsugae*” SS. Other instructions are the same as above.

**Figure 8 jof-10-00769-f008:**
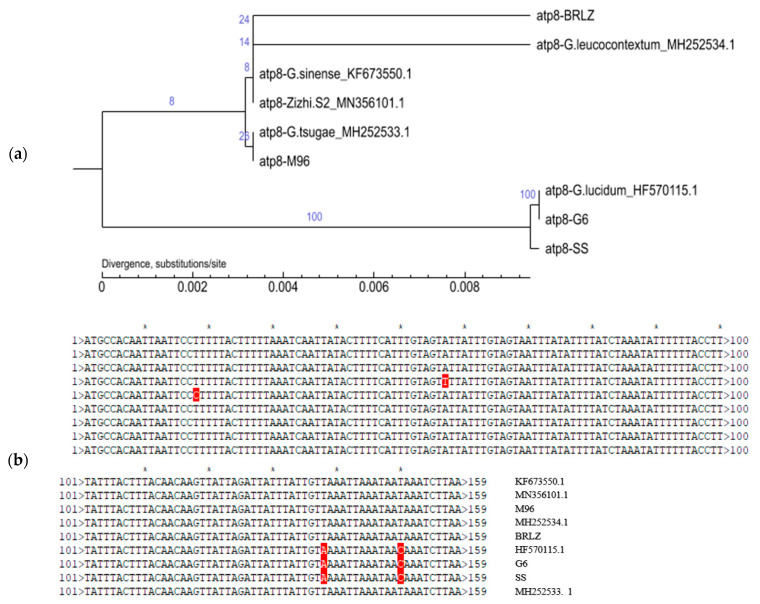
Phylogenetic tree (**a**) and sequence alignment (**b**) of *atp*8 genes G6, SS and HF570115.1 were completely consistent. BRLZ, G6, SS and MZ96 represent *G*. *leucocontextum* H4, *G. lucidum* G6, *G. tsugae* SS and *G. sinense* MZ96, respectively.

**Figure 9 jof-10-00769-f009:**
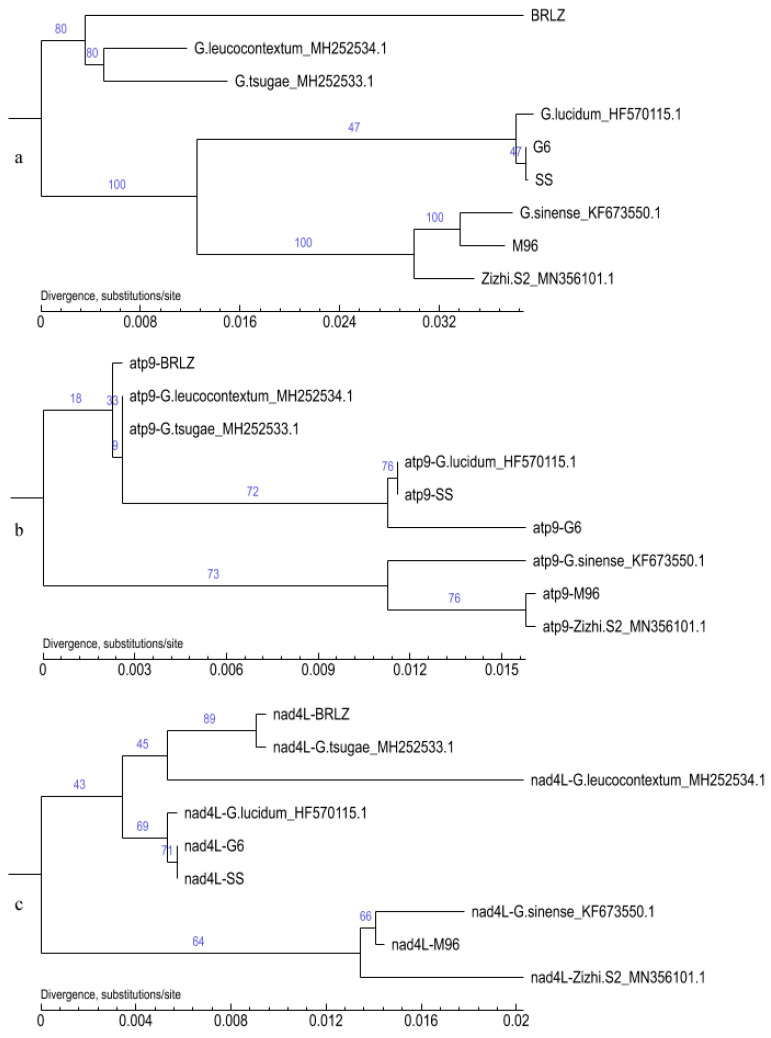
Phylogenetic trees constructed from the 15 PCGs (**a**), *atp*9 (**b**) and *nad*4L (**c**). Among those, BRLZ, G6, SS and MZ96 represent the mitogenome of *G*. *leucocontextum* H4, *G. lucidum* G6, *G. tsugae* SS and *G. sinense* MZ96, respectively.

**Figure 10 jof-10-00769-f010:**
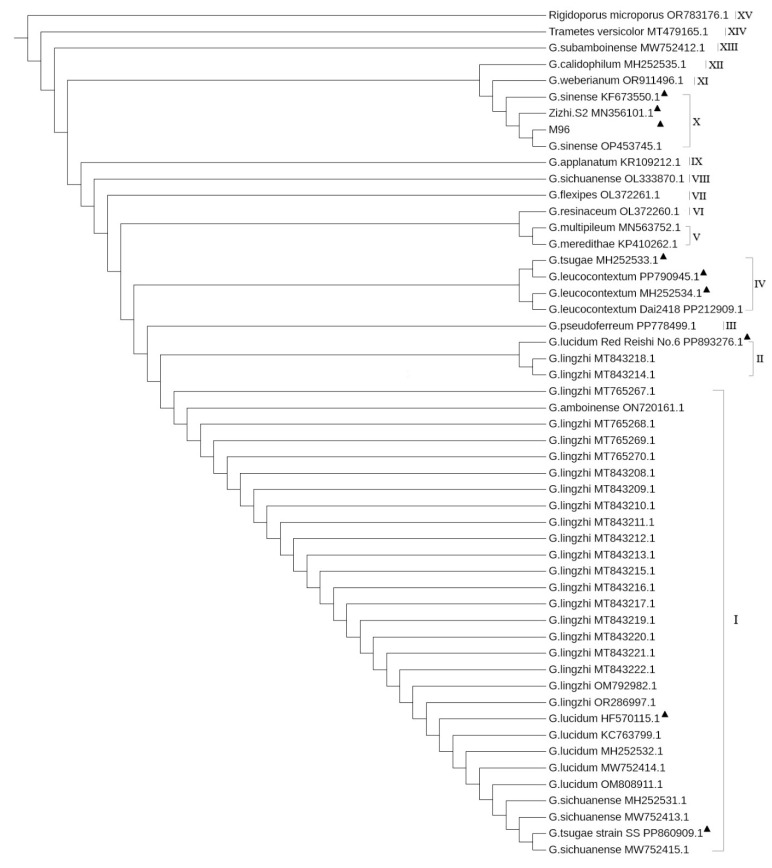
Phylogenetic tree constructed from the *atp*9 gene sequences of 52 samples. There are nine samples highlighted with **▲**, including newly completed four mitogenomes, which are used for the earlier analysis in this study. Among those, MZ96 represents *G. sinense* MZ96 (GenBank: PQ301463.1, release date: PLN 07-OCT-2024).

**Figure 11 jof-10-00769-f011:**
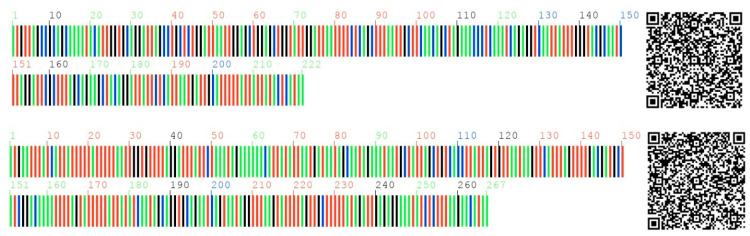
DNA barcode and QR code of *G. lucidum* strain Red Reizhi No.6 (abb. G6). The four bases of ATGC are expressed in four colors, i.e., green, red, black and blue, respectively.

**Table 1 jof-10-00769-t001:** Assembly results of mitogenomes and GC-Depth evaluation.

Genome	Seq Type	Total Number	Assembly Length (bp)	Covered Length (bp)	Coverage Rate (%)	Gap (bp)	GC Content (%)	Depth	Reads Usage Percent (%)
BRLZ	Scaffold	1	104,711	104,710	100	0	26.97	3000	5.46
G6	Scaffold	1	65,881	65,880	100	0	26.61	11,000	14.84
SS	Scaffold	1	60,320	60,319	100	0	26.69	4100	5.03
MZ96	Scaffold	1	70,554	70,553	100	0	26.08	5800	8.08

Note: BRLZ, G6, SS and MZ96 represent the mitogenome of *G*. *leucocontextum* H4, *G. lucidum* G6, *G. tsugae* SS and *G. sinense* MZ96, respectively.

**Table 2 jof-10-00769-t002:** Gene composition and information statistics of the genomes.

Genome	Genome Size (bp)	Gene				Total Length (bp)	Average Length (bp)	Total Length in Genome (%)
		CDS	ORF	rRNA	tRNA			
BRLZ	104,711	67	52	2	27	59,967	895.03	57.27
G6	65,881	43	25	2	27	40,194	934.74	61.01
SS	60,320	40	23	2	27	33,249	831.23	55.12
MZ96	70,554	46	33	2	27	41,125	894.02	58.33

Note: BRLZ, G6, SS and MZ96 represent the mitogenome of *G*. *leucocontextum* H4, *G. lucidum* G6, *G. tsugae* SS and *G. sinense* MZ96, respectively.

**Table 3 jof-10-00769-t003:** Alignment statistics of mitogenomes at nucleic acid (NA) and amino acid (AA) levels.

NA Level	ID	Refs Length	Map Length	Map Num	Rate (%)		
A and D	PQ301463.1	70,502	59,673	31	84.64		
B and D	MN356101.1	85,389	70,811	31	82.93		
C1 and E	PP790945.1	104,711	61,020	40	58.28		
C2 and E	PP893276.1	88,194	32,067	29	48.67		
C3 and E	PP860909.1	60,320	31,378	32	52.02		
C1 and F	PP790945.1	104,711	39,727	32	37.94		
C2 and F	PP893276.1	65,881	60,621	7	92.02		
C3 and F	PP860909.1	60,320	60,320	10	100		
C1 and G	PP790945.1	104,711	68,925	35	65.82		
C2 and G	PP893276.1	65,881	32,454	32	49.26		
C3 and G	PP860909.1	60,320	31,384	35	52.03		
AA level	Aligned	Genes	Target (%)	Genes	Query (%)	Identity	
						Mean	Median
A and D	30	46	65.22	41	73.17	93.70	99.70
B and D	31	59	52.54	41	75.61	96.64	99.38
C1 and E	30	67	44.78	44	68.18	92.08	98.10
C2 and E	23	43	53.49	44	52.27	92.39	96.40
C3 and E	23	40	57.5	44	52.27	92.47	96.40
C1 and F	16	67	23.88	16	100	93.99	96.59
C2 and F	16	43	37.21	16	100	99.71	100
C3 and F	16	40	40	16	100	99.72	100
C1 and G	29	67	43.28	44	65.91	93.13	98.58
C2 and G	24	43	55.81	44	54.55	86.72	92.56
C3 and G	22	40	55	44	50	90.24	92.77

Note: Newly completed reference mitogenomes in this Table are coded as follows. A: M96_PQ301463.1; B: Zizhi.S2_ MN356101.1; C1: BRLZ_PP790945.1; C2: G6_PP893276.1; C3: SS_PP860909.1; D: *G. sinense*_KF673550.1_86,451 bp; E: *G. leucocontextum*_MH252534.1_88,194 bp; F: *G. lucidum*_HF570115.1_60,631 bp and G: *G. tsugae*_MH252533.1_92,511 bp.

**Table 4 jof-10-00769-t004:** Statistics of 15 core genes and *rrn*L and *rrn*S sequences of mitogenomes.

Gene	*G. sinense*			*G. leucocontextum*		*G. lucidum*		*G. tsugae*	
	KF673550.1	MN356101.1	M96 *	MH252534.1	BRLZ *	HF570115.1	G6 *	MH252533.1	SS *
atp6	774	774	774	774	774	750	774	774	774
atp8	159	159	159	159	159	159	159	159	159
atp9	222	222	222	222	222	222	222	222	222
cob	1194	1053	1137	1161	597	1158	1158	1161	1158
cox1	1587	1587	1587	1587	1587	1611	1587	1587	1587
cox2	756	756	727	756	756	765	756	756	756
cox3	810	810	684	810	810	813	810	810	810
nad1	1017	1017	1017	1017	1017	1017	1017	1017	1017
nad2	1509	1509	1509	1509	1509	1509	1509	1509	1125
nad3	360	360	360	360	1302	360	360	354	360
nad4	1365	1452	1452	1365	1452	1455	1452	1365	1452
nad4L	267	267	267	267	267	267	267	267	267
nad5	1983	1983	1983	1983	1497	1989	1983	1983	1983
nad6	612	2175	2175	648	867	612	612	657	612
rps3	1020	1020	1020	1017	1017	1020	1020	1017	1020

Note: * Newly completed mitogenomes in this study, M96, BRLZ, G6 and SS represented *G. sinense* MZ96 (GenBank: PQ301463.1), *G. leucocontextum* H4 (GenBank: PP790945.1), *G. lucidum* G6 (GenBank: PP893276.1) and *G. tsugae* SS (GenBank: PP860909.1), respectively.

## Data Availability

The four newly completed mitogenomes presented in this study and one more recently completed mitogenome have been deposited in GenBank under the accession numbers PP790945.1 (*G*. *leucocontextum* strain H4), PP893276.1 (*G. lucidum* strain Red Reizhi No.6), PP860909.1 (*G. lingzhi* strain SS), PQ301463.1 (*G. sinense* strain Minzi 96) and MN356101, respectively.

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
