# Peer review of "Comparative Mitogenomics Provides Valuable Insights for the Phylogeny and New DNA Barcodes of Ganoderma"

_jof, 2024, doi:10.3390/jof10110769_

Round 1

Reviewer 1 Report

The manuscript entitled "Comparative mitogenomics provide new insights into phylogeny of Ganoderma" compared the  mitogenomes of Ganoderma species that included 4 newly sequenced genomes and 5 publicly available genomes. They used Trametes versicolor (Polyporaceae) and Rigidoporus microporus (Meripilaceae) as outgroups. Overall the manuscript is well written and scientifically important. However, my main concern is that the title of manuscript suggest that the manuscript will increase our knowledge of the phylogeny of Ganoderma, while I think it is rather to investigating the potential of using mitogenomes for phylogenetic studies? In the conclusion the authors stated "Utilizing collinearity and phylogenetic analyses, the atp9 and nad4L genes were identified as robust markers for novel DNA barcodes". 

1. The title of the manuscript stated that they will use mitogenomics to increase our knowledge about the phylogeny of Ganoderma species. For this reason I think that a paragraph in the introduction should explain why this (phylogeny) is needed.

2. Please add taxa information in Table 1.

3. In the introduction there are 22 whole genomes of Ganoderma species are available (Line 56). How many mitogenomes are available? 

4. In Table S4 "Total 49 mitogenomes of Ganoderma and two outgroups used for validation". Also, "32 mitogenomes in the NCBI data-base, including six records of G. lucidum, four records of G. sichuanense and 22 records of 515 G. lingzhi (Lines 514-516). This needs to be explained in detail in the Material and Methods, not the Discussion.

5. For the single gene trees, I do not understand why the authors did not include more taxa that should be available on NCBI? 

6. The authors can consider using SANS serif (alignment-free, whole-genome-based phylogeny, https://gitlab.ub.uni-bielefeld.de/gi/sans).

7. If possible that the authors can explain more clearly in the Results section the difference between species.

1. Please add taxa information in Table 1.

Author Response

Dear reviewer:

Thank you very much for encourage that this manuscript is well written and scientifically important. Your comments are most helped for us to revise and improve the previous manuscript. And all the corresponding revisions/corrections highlighted/in track changes in the re-submitted files.

  1. Questions for General Evaluation

Q1.  Does the title describe the article's topic with sufficient precision?

Response: The title of revised manuscript (jof-3234881-R1) is slightly changed “Comparative mitogenomics brings insights into the phylogeny and DNA barcoding of Ganoderma”, which maybe more precisely describe the article's topic.

Q2. Does the introduction provide a comprehensive yet concise overview about the state of knowledge in the area of research?

Response: 1. We would like to use a slightly changed-title” Comparative mitogenomics brings new perspectives for the phylogeny and novel DNA barcodes of Ganoderma”. It is expected that this study will bring about more or better selection (gene sequence) for the phylogeny and the construction of DNA barcodes. 2. About “How many mitogenomes are available?”. There are actually 49 mitogenomes of Ganoderma. (See the second paragraph of the introduction, this paragraph also explains how the data came out (see Line 64-74 for details)(Line 74-84 in revised manuscript)

Q3. Is the research design appropriate and are the methods adequately described?

Response: 1. About “This needs to be explained in detail in the Material and Methods, not the Discussion”. There are explained in the Material and Methods (see 2.7 for details, Line 162-169). (Line 200-211, in revised manuscript)

  1. About “the single gene trees, not include more taxa”. This study focuses on species of Ganoderma, especially those widely cultivated and utilized. Mitogenomic polygenic sequences were used to construct the phylogeny, including 15 core genes (14 PCGs and rps3), mtSSU and mtLSU in the mitogenomes. Therefore, we completed four mitogenomes of cultivated Ganoderma and searched the complete mitogenome sequences in the database for comparative analysis.

Of course, there are many taxa of Ganoderma registered incomplete gene sequences, at least 622 records accessed in the NCBI database (on 2024/10/16). For example, total 369 records of Ganoderma spp. mitochondrial small subunit ribosomal RNA gene(mt-SSU), partial sequence. Most are only about 493-516 bp, a few more than 1000 bp (such as GenBank: DQ661912.1_1,398 bp, GenBank: AH012389.2_1,552 bp, etc.), but all by Sanger dideoxy sequencing. Similarly, this is the case for mitochondrial ATP synthase subunit 6 (atp6) gene, partial cds (such as GenBank: AB368000.1_525 bp, GenBank: EU339253.1_630 bp, etc.).

  1. Thank you for your mentioned SANS serif (https://gitlab.ub.uni-bielefeld.de/gi/sans), we will seriously consider using this software in future studies.

Q4. About “explain more clearly in the Results section the difference between species”

Response: Thank you for your suggestion, some appropriate modifications and supplements were made in the Results section to explain the differences between species more clearly.

  1. II. Comments and Suggestions for Authors

Thank you. Your comments and suggestions are very pertinent. In the revision, we would like to adjust the paper title as “Comparative mitogenomics brings new perspectives for the phylogeny and novel DNA barcodes of Ganoderma

Comments 1: 1. The title of the manuscript stated that they will use mitogenomics to increase our knowledge about the phylogeny of Ganoderma species. For this reason, I think that a paragraph in the introduction should explain why this (phylogeny) is needed.

Response: Thank you very much. We also have this consideration and and two paragraphs has been added into different positions in the Introduction (the second and the last paragraph), respectively.

Comments 2. Please add taxa information in Table 1.

Response: Ok, has been added up.

Comments 3. In the introduction there are 22 whole genomes of Ganoderma species are available (Line 56). How many mitogenomes are available? 

Response: Ok, it was described in the next paragraph, and updated the accessed date (Oct 7, 2024)

Comments 4. In Table S4 "Total 49 mitogenomes of Ganoderma and two outgroups used for validation". Also, "32 mitogenomes in the NCBI data-base, including six records of G. lucidum, four records of G. sichuanense and 22 records of 515 G. lingzhi (Lines 514-516). This needs to be explained in detail in the Material and Methods, not the Discussion.

Response: There is an error, has been revised accordingly. The point is on the second half of this sentence, which was discussing an interesting analysis result, maybe as a reference for future studies (Line 512-514 in revised manuscript).

Comments 5.  For the single gene trees, I do not understand why the authors did not include more taxa that should be available on NCBI? 

Response: Ok, has been added up.

Comments 6. The authors can consider using SANS serif (alignment-free, whole-genome-based phylogeny, https://gitlab.ub.uni-bielefeld.de/gi/sans).

Response: Ok, has been added up.

Comments 7. If possible that the authors can explain more clearly in the Results section the difference between species.

Response: Ok, has been added up.

Detail comments

  1. Please add taxa information in Table 1.

Response: Ok, the taxa information of four sequenced strains have been added.

Reviewer 2 Report

Ti-Qiang and colleagues sequenced the mitogenomes of four Ganoderma strains, upon which multiple genetic and phylogenetic analyses were performed. However, for me, it is really hard to see the goal of this research and the significance of this work. Therefore, I would not suggest this manuscript to be published. My specific questions and comments on this work are followed:

Abstract should be improved. What is Ganoderma? Why do we care? What is mitogenome? Why is it important to study the mitogenome of Ganoderma? What is known and unknown? What is the major question that the authors would like to answer in this work? The same goes to Introduction.

In section 3.1, four strains sequenced were named differently from those in Table 1. The same issue happens throughout the entire manuscript, especially on H4/BRLZ. Please check and make them consistent.

The resolution of Figure 2 is really low, making it hard to read. Please provide higher res figure if possible.

In section 3.3, it is stated that ‘GO provided the best annotations’. How to define ‘best annotations’? I appreciate it that the authors provide the raw data regarding this conclusion (Supplementary III), but it is truly hard to interpret for readers that do not have sequencing background.

In the last paragraph of section 3.3 (line 287-296), it is reported that some DNA polymerases identified in the mitogenomes have 43 aa and 190 aa, which seem to be extremely small even for mitochondrial DNA polymerases. Can the authors be 100% sure that these annotations are valid? If so, can the authors provide possible explanations for this observation?

Section 3.4 and its corresponding figures need to be improved. What is collinearity analysis? Why did the authors decide to do this? What is the advantage of performing this analysis in this research? What is the rationale of including strain Zizhi S2 in this assay? What is ‘scaffold 1’ in Figure 4, 5, 6 7?

The scientific logic of Section 3.5 is hard to follow. How can genes with the longest and shortest lengths be excluded from an analysis because of inconsistency with the result from another analysis?

Figure S7 does not show amino acid identity between QPF23604.1 and XBQ63844.1.

Figure S9 and 11 do not exist in the manuscript submitted.

The conclusion of section 3.8.3 is confusing. What assay was performed for the validation of atp9 and nad4L as DNA barcodes? These barcodes were directly used to be compared to their potential homologs. Results of the genetic comparisons do not validate the authenticity of the barcodes.

Author Response

Dear reviewer:

Thank you very much for encourage that this manuscript is well written and scientifically important. Your comments are most helped for us to revise and improve the previous manuscript. And all the corresponding revisions/corrections highlighted/in track changes in the re-submitted files.

  1. Questions for General Evaluation

Q1.  Does the title describe the article's topic with sufficient precision? This work provides 'new' mitogenomes, but not 'new' insights.

Response: Thanks. The title of revised manuscript is slightly changed “Comparative mitogenomics brings insights into the phylogeny and DNA barcoding of Ganoderma”, which maybe more precisely describe the article's topic.

Q2. Does the introduction provide a comprehensive yet concise overview about the state of knowledge in the area of research? The introduction lacks a background brief on the topic.

Response: Thanks. We also have this consideration and two paragraphs has been added into different positions in the Introduction (the second and the last paragraph), respectively.

Q4. Specific comments are followed.

Response: Thanks. According to your followed specific comments, we made appropriate modifications and supplements.

Q7. Moderate editing of English language required.

Response: Thanks. This manuscript has been revised in order to improve the quality of English language

  1. II. Comments and Suggestions for Authors

It is really hard to see the goal of this research and the significance of this work.  I would not suggest this manuscript to be published.

Response: Thank you for your review and encouragement.

Sorry, the Abstract and Introduction have not written properly. Now, we made careful revisions to clearly state the goal of the research and the significance of this work.

Detail comments

Comments 1: Abstract and Introduction should be improved.

Response: Thanks, the Abstract have been revised, and two paragraphs were supplemented to Introduction

Comments 2: In section 3.1, four strains sequenced were named differently from those in Table 1. The same issue happens throughout the entire manuscript, especially on H4/BRLZ. Please check and make them consistent.

Response: Ok, we have made supplementary explanations in the corresponding places, and keep them consistent throughout the article. The samples (four sequenced strains) were labeled with corresponding abbreviation for bioinformatics analysis, such as BRLZ for the G. leucocontextum strain and H4 for the original name of this strain. Such description was noted in each Table and Figure as supplementary.

Comments 3: The resolution of Figure 2 is really low, making it hard to read. Please provide higher res figure if possible.

Response: Ok. In revised manuscript, the original graph (about 25.7M of pixel size) was used in Figure 2, which can be display more clearly and more readable when it enlarged.

Comments 4: In section 3.3, it is stated that ‘GO provided the best annotations. How to define ‘best annotations? I appreciate it that the authors provide the raw data regarding this conclusion (Supplementary III), but it is truly hard to interpret for readers that do not have sequencing background.

Response: Ok, it has been corrected and the corresponding notes are added in Table S5

Comments 4: In the last paragraph of section 3.3 (line 287-296), it is reported that some DNA polymerases identified in the mitogenomes have 43 aa and 190 aa, which seem to be extremely small even for mitochondrial DNA polymerases. Can the authors be 100% sure that these annotations are valid? If so, can the authors provide possible explanations for this observation?

Response: This interesting annotation result (the dpo gene contains 2~3 exons, encodes 43 aa, 99 aa and 728 aa) has been observed, but without further analysis and discussion. But it has little correlation to the subject of this study, so we decided to delete this sentence.

Comments 5: Section 3.4 and its corresponding figures need to be improved. What is collinearity analysis? Why did the authors decide to do this? What is the advantage of performing this analysis in this research? What is the rationale of including strain Zizhi S2 in this assay? What is ‘scaffold 1’ in Figure 4, 5, 6 7?

Response: We made appropriate modifications to Figure 4,5,6 and 7 which generated by the collinearity analysis software, and the ‘scaffold 1’ (mitogenome assembly) were replaced with their GenBank accession number (MZ96: PQ301463.1; G6: PP860909.1; SS: PP893276.1).

Collinearity analysis is an essential analytical strategy in comparative genomes because it allows the analysis of molecular evolutionary events at large and small scales between species or different strains within species. According to the collinearity analysis results in this study, it was confirmed that the mitogenomes of the same species should exhibited more than 82% identity at nucleic acid level. For example, 84.64% identity between G. sinense M96 and G. sinense_KF673550.1, 82.93% identity between Zizhi.S2_MN356101.1 and G. sinense_KF673550.1 (Table 3), and 92.02% identity between G. lucidum Red Reishi No.6_PP893276.1 and G. lucidum_HF570115.1 (Table 3). It is noteworthy that the collinearity analysis of the mitogenomes of ‘G. tsugae’ SS_PP860909.1 and G. lucidum_HF570115.1 exhibited a 100% identity at the nucleic acid level. This finding implies that the cultivated strain SS may represent a distinct species from G. tsugae, despite its initial classification as G. tsugae base on morphological taxonomy.

Recently, our previously sequenced strain ‘Zizhi S2’ was emended as G. sinense based on comparative analysis whole genome (nuclear and mitochondrial), see Reference 54 (Genomics, Sept.2024). It is more worthy of comparative analysis with the newly completed mitogenome of G. sinense MZ96 in this study.

Scaffold 1’ mean the mitogenome assembly, which have not submitted to NCBI when the collinearity analysis was performing.  And GenBank PQ301463.1 was just released on PLN 07-OCT-2024)

Comments 6: The scientific logic of Section 3.5 is hard to follow. How can genes with the longest and shortest lengths be excluded from an analysis because of inconsistency with the result from another analysis?

Response: How to select the most reliable atp9 and nad4L from the 15 candidate genes, we fully refer the relevant research literature, and take into consideration both the generation of DNA barcodes and the construction of phylogenetic trees.

As candidate genes for DNA barcoding, the more suitable sequence lengths were 200 – 400bp. If the candidate gene sequence is long, it is not good for researchers to design primers and amplify sequencing to obtain the complete gene sequence; If the candidate gene sequence is short, the single nucleotide polymorphism of homologous genes between different species or strains is small and not valuable.

     In this study, these two genes (atp9 and nad4L) have been identified as reliable marker of new DNA barcodes, offering valuable insights and contributing significantly to understanding the evolutionary relationships and phylogeny of Ganoderma genus and even the Ganodermataceae family

Comments 7: Figure S7 does not show amino acid identity between QPF23604.1 and XBQ63844.1.

Response: Sorry, the Figure S7 should be corrected as Figure S9 (a: atp9 gene; b:  nad4L gene)

Comments 8: Figure S9 and 11 do not exist in the manuscript submitted.

Response: Sorry, the Figure S9 was wrong labeling as Figure S10 (not exist), and Figure 11 was shown in Line 353-356 (DNA barcode and QR code) has been corrected (Line 433-436 in the revised manuscript).

Comments 9: The conclusion of section 3.8.3 is confusing. What assay was performed for the validation of atp9 and nad4L as DNA barcodes? These barcodes were directly used to be compared to their potential homologs. Results of the genetic comparisons do not validate the authenticity of the barcodes.

Response: Ok, we decided to delete the Section 3.8.3, which seems confusing.

In fact, this Section describes how to use the atp9 and nad4L as DNA barcodes, to mine homologous genes from an unannotated genomic assembly (GenBank: GCA_024305745.1) and most likely mitogenome sequence (GenBank: JAHRBS010000017.1), one atp9 gene (82762-82541bp) and two nad4L genes (112036-111770 bp, 19602-19336 bp) were successfully obtained.

Reviewer 3 Report

I believe the article is devoted to a relevant topic, modern methods were used, and interesting data was obtained.

It is recommended to make the axis labels in the graphs in Figure 3 and Figure 2 more readable. I consider Figure 1a to be redundant. It would have been sufficient to show only a few mushroom fruiting bodies.

Author Response

Dear reviewer:

      Thank you very much for your affirmation. Based on the comments of all reviewers, we have slightly changed the title to “Comparative mitogenomics brings insights into the phylogeny and DNA barcoding of Ganoderma”, and hope to get your understanding and support.

  1. II. Comments and Suggestions for Authors

Major comments

     I believe the article is devoted to a relevant topic, modern methods were used, and interesting data was obtained.

Detail comments

It is recommended to make the axis labels in the graphs in Figure 3 and Figure 2 more readable. I consider Figure 1a to be redundant. It would have been sufficient to show only a few mushroom fruiting bodies.

Response: Thanks for your advice. After properly edited, now the Figure 1A only show a few fruiting bodies. It is necessary to use this graph to show that the cultivated fruiting bodies of strain SS are very similar to G. tsugae in morphological characteristics, and regarded as "G. tsugae " locally cultivated in Changbai Mountain (Jilin Province, China).

In revised manuscript, the original graphs with PNG or JPG format were used in Figure 2 and Figure 3, with 25.7M and 11.6 M of pixel size, respectively. Now, the axis labels of Figure 3 and Figure 2 can be display more clearly and more readable when it enlarged.

Such as the graph c in Figure 2, can be enlarged to 105.82 cm long and 105.82 cm wide. In fact, the actual size of this part is as follows:

   So does the graph a in Figure 3.

The original graph in Figure 3 can be enlarged to actual size (71.65cm long and 70.49cm wide). In fact, the actual size of this part is as follows:

Round 2

Reviewer 2 Report

All of my previous comments have been addressed. Thanks to the authors for their efforts in improving this manuscript.

All of my previous comments have been addressed. Thanks to the authors for their efforts in improving this manuscript.